# Whether Renal Pathology Is an Independent Predictor for End-Stage Renal Disease in Diabetic Kidney Disease Patients with Nephrotic Range Proteinuria: A Biopsy-Based Study

**DOI:** 10.3390/jcm12010088

**Published:** 2022-12-22

**Authors:** Tingli Wang, Junlin Zhang, Yiting Wang, Lijun Zhao, Yucheng Wu, Honghong Ren, Yutong Zou, Rui Zhang, Huan Xu, Zhonglin Chai, Mark E. Cooper, Jie Zhang, Fang Liu

**Affiliations:** 1Division of Nephrology, West China Hospital of Sichuan University, Chengdu 610041, China; 2Laboratory of Diabetic Kidney Disease, Centre of Diabetes and Metabolism Research, West China Hospital of Sichuan University, Chengdu 610041, China; 3Division of Pathology, West China Hospital of Sichuan University, Chengdu 610041, China; 4Department of Diabetes, Central Clinical School, Monash University, Melbourne, VIC 3004, Australia; 5Laboratory of Transplant Engineering and Immunology, Ministry of Health, Regenerative Medicine Research Center, Chengdu 610041, China

**Keywords:** diabetic kidney diseases, end-stage renal disease, nephrotic range proteinuria, renal biopsy

## Abstract

Aims: To investigate whether renal pathology is an independent predictor for end-stage renal disease (ESRD) in diabetic kidney diseases (DKD) with nephrotic range proteinuria. Methods: A total of 199 DKD patients with nephrotic range proteinuria underwent renal biopsy and were divided into an ESRD group and a non-ESRD group. A Kaplan–Meier analysis was used to compare renal survival rate, and univariate and multivariate Cox proportional hazard analyses were used to determine the predictors of the ESRD. Results: The mean age of included patients was 51.49 ± 9.12 years and 113 patients (56.8%) progressed to ESRD. The median follow-up period was 16 (12–28) months. The glomerular pathology class III is the most common type (54.3%). In the Kaplan–Meier analysis, compared with patients without ESRD, patients with ESRD had a longer duration of diabetes (≥6 years), lower eGFR (<60 mL/min/1.73 m^2^), lower albumin (<30 g/L), lower hemoglobin (<120 g/L), and a higher grade of glomerular stage (class III + IV vs. class I + II) (*p* < 0.05). The hemoglobin and e-GFR, but not the histopathological damage, were significantly associated with a higher risk of ESRD in both the univariate and multivariate Cox analyses. Conclusions: In patients with diabetic kidney disease characterized by nephrotic range proteinuria, histopathological damage (glomerular alterations, interstitial fibrosis and tubular atrophy (IFTA), interstitial inflammation, and arteriolar hyalinosis) is not associated with poor renal outcomes, but hemoglobin and e-GFR could predict poor renal outcomes.

## 1. Introduction

Diabetes is one of the fastest-growing health challenges of the 21st century. The latest edition of the International Diabetes Federation (IDF) Diabetes Atlas shows that 537 million adults are currently living with diabetes. China is anticipated to have the largest number of adults with diabetes until 2030 [1]. Diabetic kidney disease (DKD), a known microvascular complication of type 1 and type 2 diabetes, is the most common cause of chronic kidney disease (CKD) and end-stage renal disease (ESRD). In developed countries and China, DKD accounts for approximately 50% and 16.4% of all cases of ESRD, respectively [2,3,4]. Early diagnosis and treatment could play a key role in preventing ESRD.

Clinicians pay attention to diabetic patients with albuminuria or a decreased glomerular filtration rate (GFR), although this does not confirm the diagnosis of DKD. Other clinical indicators, as well as renal biopsy where necessary, are required to make a definite DKD diagnosis. Current guidelines [5,6,7] recommend that an increase in urine albumin/creatinine ratio (UACR) and decrease in estimated glomerular filtration rate (e-GFR) in diabetes patients could be an important predictor of the occurrence and development of DKD. Recent studies have reported that the remission of nephrotic-range albuminuria (>2.5 g/24 h) reduces the risk of end-stage renal disease and improves survival in type 2 diabetic patients [8]. However, there are few studies and reports on the pathological indicators related to the prognosis of kidney disease in these DKD patients with nephroic range proteinuria (proteinuria ≥ 3.5 g/24 h). Therefore, we retrospectively investigated 199 (DKD) patients diagnosed by renal biopsy with nephrotic range proteinuria in West China Hospital, focusing primarily on the occurrence of ESRD and other parameters such as clinical, histologic, and laboratory indicators.

## 2. Materials and Methods

### 2.1. Patient Inclusion and Exclusion Criteria

A total of 398 patients with biopsy-proven DKD in West China Hospital of Sichuan University from January 2008 to November 2018 were reviewed, and 199 patients with nephrotic range proteinuria (proteinuria ≥ 3.5 g/24 h) were eligible (Figure 1). The general indications for renal biopsy in this study were type 2 diabetes mellitus (T2DM) patients with renal damage who lacked absolute contraindications, especially T2DM patients without diabetic retinopathy (DR) or T2DM patients with obvious glomerular hematuria and/or a short diabetic duration, or with sudden-onset overt proteinuria. The diagnosis of T2DM and DKD followed the American Diabetes Association (ADA) and the Renal Pathology Society (RPS), respectively [9,10]. The exclusion criteria of patients included a follow-up time of less than 1 year, a lack of information regarding proteinuria, progression to ESRD before renal biopsy, a lack of follow-up, and death. In this study, all renal biopsies were performed with the consent of the patient. The study protocol was reviewed and approved by the hospital ethics committee. Only patients who consented to this study were included and analyzed in the final analysis.

The patients had regular follow-ups and information regarding their proteinuria and renal function was collected. Renal outcome in this study was defined by the progression to ESRD, which was indicated by e-GFR < 15 mL/min/1.73 m^2^ or the initiation of renal replacement therapy and was the end point of our study. Renal survival was assessed by considering whether the patient had reached the end point of our study at the follow-up. Patients not reaching the end point were evaluated using the medical records of their last follow-up visit. Using this information, the patients in this study were divided into two groups (non-ESRD group and ESRD group).

### 2.2. Clinical, Laboratory, and Pathologic Characteristics

The baseline data were collected when patients were hospitalized for renal biopsy. Clinical characteristics were collected at the time of renal biopsy, including age, gender, duration of diabetes, systolic and diastolic blood pressures, blood glucose, HbA1c, 24 h urinary protein, serum creatinine (umol/L), e-GFR (mL/min/1.73 m^2^), serum albumin, total cholesterol, triglyceride levels, and hemoglobin. Information regarding medication history, especially the use of insulin, statin and renin-angiotensin-aldosterone system (RAAS) inhibitors, and angiotensin-converting enzyme (ACE) inhibitors or angiotensin II receptor blockers (ARBs), was also collected.

For each biopsy specimen, light microscopy, immunofluorescence, and electron microscopy were routinely performed and evaluated by the same group of pathologists. The pathological classifications of glomerular alterations, interstitial fibrosis and tubular atrophy (IFTA), interstitial inflammation, and arteriolar hyalinosis were categorized based on the pathologic classification of the RPS [10].

### 2.3. Statistical Analysis

Patient characteristics that differed between the two groups were compared using Student’s *t*-test or Wilcoxon signed-rank test for continuous variables and, where appropriate, the Chi squared test or Fisher’s exact test for categorical variables. For continuous variables, data were presented as mean ± standard deviation or median and interquartile range, and categorical variables as number and percentages. Correlations between the histopathological and clinical findings were analyzed by Spearman’s correlation analysis. Kaplan-Meier and univariate and multivariate cox regression analyses were performed to assess the association of included variables with the presence of ESRD. For all the tests, *p* values < 0.05 were considered statistically significant. All statistical analyses were performed using R software (version 3.6.2, Ross Ihaka and Robert Gentleman, Auckland, New Zealand) and SPSS (IBM SPSS 26.0, SPSS Inc., Chicago, IL, USA).

## 3. Results

### 3.1. Baseline Clinical and Pathologic Characteristics

Of the 199 patients enrolled in the study, the mean age was 51.49 ± 9.12 years, and 58 (29.1%) patients were male (Table 1). The median duration of DM was 96 (36–132) months and the median follow-up period was 16 (12–28) months. The mean baseline serum creatinine level was 139 (97–187) mmol/L, the mean e-GFR was 51.31 (36.15–81.54) mL/min/1.73 m^2^, and the mean 24 h proteinuria was 6.78 (4.8–9.75) g/day. A total of 113 patients (56.8%) progressed to ESRD during follow-up. The mean patient age of ESRD patients was 50.9 ± 8.6 years old and males made up 27.4%. Compared with the non-ESRD group, patients in the ESRD group had higher levels of serum creatinine, Cys-C, and BUN, but lower levels of albumin, hemoglobin, HbA1c, e-GFR, and, significantly, calcium (*p* < 0.05) (Table 1).

Based on the glomerular classification [10], the glomerular pathologic class III is the most common type (54.3%). Among the patients who progressed to ESRD, the percentage of classes I, IIa, IIb, III, and IV was 0, 25, 52.2, 64.8, and 63.2%, respectively. Other factors including interstitial fibrosis and tubular atrophy (IFTA), interstitial inflammation, and arteriolar hyalinosis scores were also included (Table 2). Patients in the ESRD group had more serious glomerular damage (*p* < 0.001), IFTA lesions (*p* = 0.047), and interstitial inflammation (*p* = 0.003) compared to the non-ESRD group. After analyzing the results of immunofluorescence, patients in the ESRD group were found to have more IgG (*p* = 0.010), IgM (*p* = 0.025), C3 (*p* = 0.003), C4 (*p* = 0.049), and C1q (*p* = 0.015) deposited than patients in the non-ESRD group (Table 3). However, there was no significant difference between the two groups with respect to IgA deposition (*p* = 0.594).

### 3.2. Associations between the Histopathological and Clinical Findings

The correlations between the histopathological and clinical findings are illustrated in Table 4. ESRD showed a significant positive correlation with the glomerular class (r = 0.228, *p* < 0.001) and interstitial inflammation scores (r = 0.219, *p* < 0.001), but no relationship with the IFTA scores and arteriolar hyalinosis scores. Moreover, ESRD had an inverse correlation with albumin level (r = −0.253, *p* < 0.001) and e-GFR (r = −0.359, *p* < 0.001) and hemoglobin (r = −0.437, *p* < 0.001) when adjusting for the baseline age and gender. Interestingly, when proteinuria was greater than 3.5 g/24 h, there was no correlation with end-stage renal disease.

### 3.3. Clinical and Pathological Risk Factors for ESRD

As shown in Table 5, the univariate Cox analysis demonstrated that the diabetes duration (*p* = 0.042), e-GFR (*p* < 0.001), albumin (*p* < 0.001), Hb (*p* < 0.001), proteinuria (*p* = 0.006), use of ACEI/ARB (*p* = 0.001), use of Insulin (*p* = 0.004), glomerular damage (*p* < 0.001), IFTA lesions (*p* = 0.002), and interstitial inflammation (*p* < 0.001) were associated with a higher risk for renal dysfunction.

The variates with significant univariate Cox analysis results and arteriolar hyalinosis were included in the Kaplan-Meier analysis (Figure 2A–J). Patients with a longer diabetes duration (≥6 years (72 months)), lower eGFR (<60 mL/min/1.73 m^2^), lower albumin (<30 g/L), lower hemoglobin (<120 g/L), and a higher grade of glomerular damage (class III + IV vs. class I + II) achieved a worse prognosis significantly (*p* < 0.05). Patients treated with ACEI/ARB or insulin seemed to have better prognosis (*p* < 0.05).

The multivariate cox analysis (Figure 3) was conducted with arteriolar hyalinosis and the significant variates in univariate analysis. Both e-GFR (HR 0.979; 95% CI, 0.968–0.991, *p* < 0.001) and Hb (HR 0.986; 95% CI, 0.975–0.996, *p* = 0.008) remained independent risk factors.

## 4. Discussion

In the present study, the prognostic impact of clinical, pathological parameters was analyzed in 199 cases of DKD diagnosed by renal biopsy with nephrotic range proteinuria. Multivariate analysis showed that lower hemoglobin and lower e-GFR were significantly associated with poorer renal outcomes, while the duration of diabetes, albumin level, proteinuria, use of ACEI/ARB, use of insulin, and histopathological damage were not associated with adverse renal outcomes.

It is worth mentioning that there are several important negative results in our study. Firstly, in our study, renal pathology was not an independent predictor of ESRD in DKD patients with nephrotic range proteinuria. The reason could be that most patients in our cohort were in the glomerular pathologic class III (54.27%). However, pathological stage I, IIA, IIb, and IV accounted for only 1%, 14.07%, 11.56%, and 19.1%, respectively. Moreover, in clinical practice, most of the class-IV patients have poor renal function combined with anemia, hypertension, and other complications related to DKD. Therefore, renal biopsy will not be performed in such patients because of the greater risk of bleeding. This might be the reason why glomerular pathologic class III had the highest percentage of entering ESRD in our study. In this study, although the pathological changes were not statistically significant in the multivariate analysis for renal prognosis, these changes (glomerular class and interstitial inflammation) were positively correlated with the risk of renal failure in patients with DKD, which shows the clinical significance of pathologic indicators in the diagnosis and prognosis of DKD. Meanwhile, DKD has various phenotypes with different clinical manifestations and pathologies. Especially in type 2 diabetes, weaker associations were observed between histopathological and clinical findings, probably due to heterogeneity in the trajectories of GFR and albuminuria [11]. Although renal biopsy is not necessary for DM patients to be diagnosed with DKD clinically, it is worth noting that the pathological results drawn from renal biopsy of about 1/5 DM patients are inconsistent with clinical diagnosis [12]. Moreover, there are disputes in the relationship between the pathology and prognosis of diabetic nephropathy and data are limited regarding pathologic phenotypes of DKD [4,13,14,15,16]. The ability of pathologic changes to predict the outcome also depends on the severity of disease [16]. Therefore, for patients with T2DM complicated with proteinuria or decreased e-GFR, we recommend a positive attitude towards renal biopsy after assessing the related risks. The renal biopsy is meaningful for differentiating pure DKD from NDRD because of different renal outcomes [17,18,19,20]. An early renal biopsy to clarify the pathology also has a positive effect on the prognosis of DKD [4,13].

Secondly, multivariate analysis showed that a low albumin level was not an independent risk factor for ESRD in DKD patients with nephrotic range proteinuria, a group in which 74.4% of patients exhibited serum albumin <35 g/L. Patients with hypoproteinemia would be prone to anemia, followed by anemic hypoxia and accelerating kidney damage [21]. To prevent such circumstances, RAAS inhibitors and Keto-analogs were recommended in clinical practice in treating DKD patients with nephrotic range proteinuria, since these could reduce the level of proteinuria and maintain the level of serum albumin [22,23] and delay renal function decline. Contrary to existing studies [11], some common risk factors in this study were found to be insignificant, including age, gender, smoking habits, duration of diabetes, blood glucose level, hypertension, dyslipidemia, etc. This may be due to the complex pathogenesis of DKD [24] and the difference in potential pathogenesis of various phenotypes [11].

Anemia is a common and major complication in patients with CKD and also occurs in DKD. Several studies have found that Hb is an independent risk factor for the progression of kidney disease to ESRD in type 1 and type 2 diabetes mellitus [25,26,27,28]. Lower Hb concentration in T2DM patients without clinical albuminuria may be a significant predictor of subsequent decline in GFR [29]. In our study, patients with HB ≥ 120 g/L had a significantly longer median survival time than that of the <120 g/L group. In addition, Rossing et al. [30] found that low baseline hemoglobin was significantly associated with the composite end-point risk of the doubling of serum creatinine or ESRD. DKD-related anemia developed earlier and was more severe than non-DKD-related anemia based on different mechanisms [31], such as source deficiency for RBC production, greater bleeding tendency, poor response to EPO, tubulointerstitial damage [32], chronic inflammation, etc. [31]. In a 2016 consensus stated in India [27], the early identification of anemia for treatment of DKD-related anemia is recommended to improve outcomes. Recently, erythropoiesis stimulating agents (ESAs) and iron supplementation have drawn attention [33]. Moreover, the hypoxia inducible factor proline hydroxylase inhibitor (HIF-PHI) has been approved for the treatment of anemia in patients with dialysis-dependent or non-dialysis-dependent CKD [34,35]. The use of HIF-PHI to treat DKD patients without dialysis was expected to improve both renal anemia and prognosis [36]. Our study suggests that earlier treatment of anemia might improve renal prognosis in such DKD patients with nephrotic range proteinuria.

Estimated GFR is widely accepted as not only an indicator of renal function, but also a predictor of the prognosis of DKD, aside from urinary albumin/creatinine ratio [37]. Our study suggests that e-GFR at biopsy is an independent risk factor for progression to ESRD in DKD patients with nephrotic range proteinuria. Consistent with previous studies, our study also suggests that e-GFR is an independent risk factor for renal progression in both CKD and DKD [13,38,39]. There was no positive conclusion in our study regarding the use of ACEI/ARB. However, at present, ACEI or ARB could significantly delay the progression of nephropathy and ESRD for patients with diabetes and hypertension, proteinuria or e-GFR < 60 mL/min/1.73 m^2^ [40,41,42,43]. In 2019, CREDENCE trial proved the efficacy of SGLT2 inhibitors for DKD: compared with that of the placebo group, the risk of renal composite endpoint events (ESRD, doubling of serum creatinine, and kidney-related death) was reduced by 34% in the caloglitazone group [44]. However, SGLT2 inhibitors were not used in our cohort. The use of the combination of ACEI or ARB with SGLT2 inhibitors are expected to further increase e-GFR. In addition, we should avoid the occurrence of AKI so as to slow down the decrease of e-GFR.

The study has several limitations. Firstly, this is a real-world, single-center, retrospective study with a relatively small sample size, using e-GFR to assess renal survival. Secondly, the follow-up time is relatively short and an evaluation of treatment efficacy during the follow-up was not conducted. Third, histological changes are evaluated according to a simple classification of PRS, excluding much more detailed and complicated histological changes. Other variables, including medical maturity and education level, which may affect the prognosis of DKD, were not evaluated in this study. Therefore, further study should be carried out on a larger population.

In conclusion, this study suggests that in patients with DKD complicated by nephrotic range proteinuria, lower hemoglobin and lower GFR are significantly associated with adverse renal outcomes. However, histopathological damage (glomerular alterations, interstitial fibrosis and tubular atrophy (IFTA), interstitial inflammation, and arteriolar hyalinosis) is not associated with poor renal outcomes. Moreover, the pathological features, predictive factors, and prognosis of DKD may vary with phenotype. Further studies on prospective renal biopsy are needed to investigate the structural changes of the kidney in different clinical courses of DKD, which may help us to fully understand the mechanism of DKD progression and its relationship with renal prognosis.

## Figures and Tables

**Figure 1 jcm-12-00088-f001:**
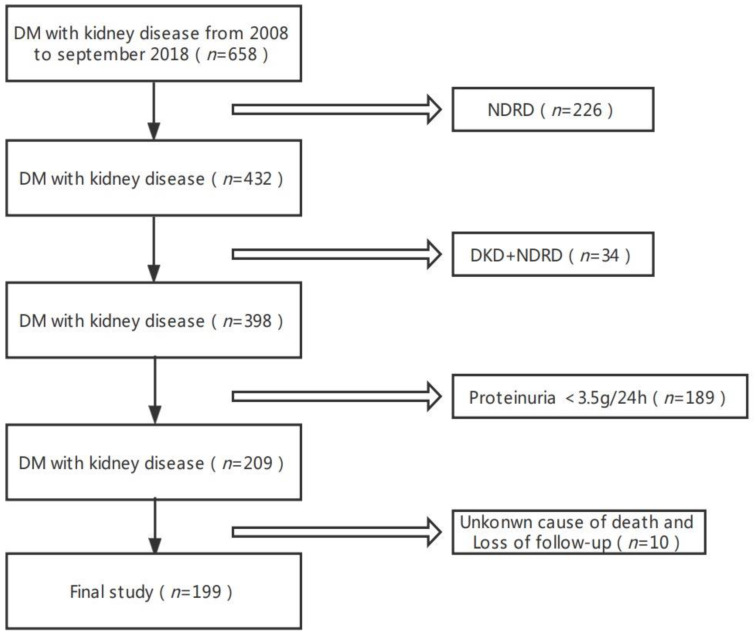
Flowchart of study participants.

**Figure 2 jcm-12-00088-f002:**
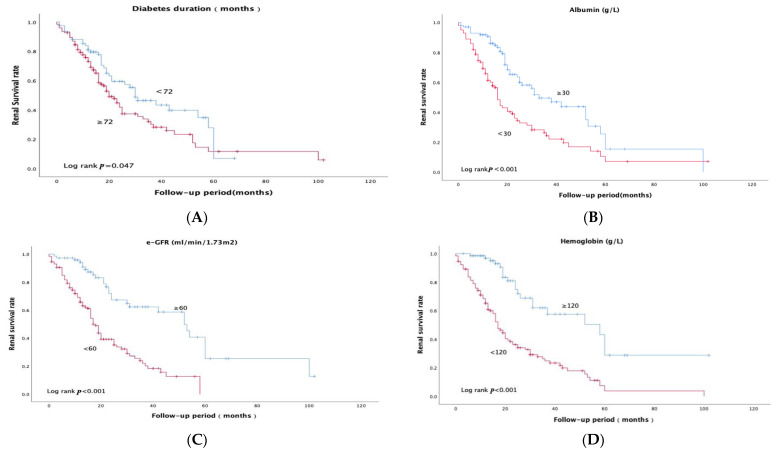
Kaplan–Meier curves of the renal survival rate in DKD patients with nephrotic range proteinuria (**A**–**J**). (**A**) Diabetes duration < 72 months group and diabetes duration ≥ 72 months group. (**B**) Albumin < 30 g/L group and albumin ≥30 g/L group. (**C**) e-GFR < 60 mL/min/1.73 m^2^ group and e-GFR ≥60 mL/min/1.73 m^2^ group. (**D**) Hemoglobin < 120 g/L group and hemoglobin ≥ 120 g/L group. (**E**) Patients treated with ACEI/ARB group and without ACEI/ARB group. (**F**) Patients treated with insulin group and without insulin group. (**G**) Glomerular damage (glomerular class I + II group and class III + IV group). (**H**) Interstitial fibrosis and tubular atrophy (IFTA), <25% group and ≥25% group. (**I**) Interstitial inflammation (0 + 1 group and 2 group). (**J**) Arteriolar hyalinosis (0 + 1 group and 2 group). Interstitial inflammation (0, absent; 1, infiltration only in relation to IFTA; 2, infiltration in areas without IFTA) arteriolar hyalinosis (0, absent; 1, at least one area of arteriolar hyalinosis; 2, more than one area of arteriolar hyalinosis).

**Figure 3 jcm-12-00088-f003:**
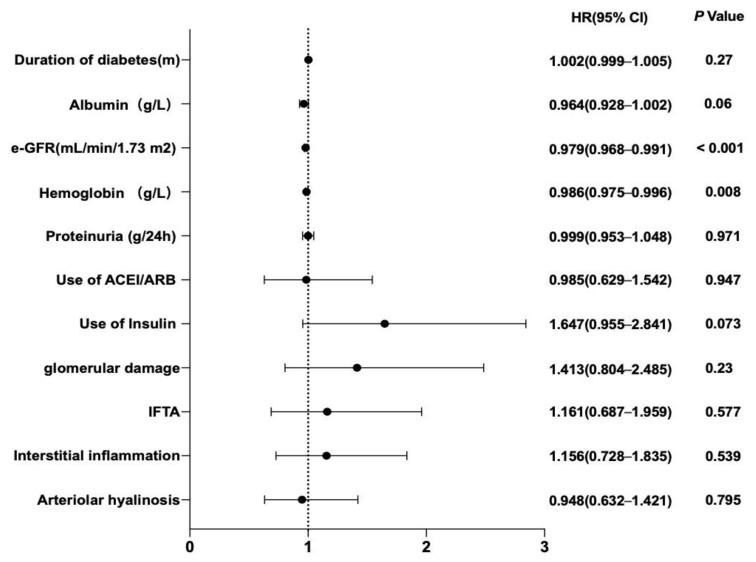
Risk factors for ESRD identified by multivariate Cox analysis in DKD patients with nephrotic range proteinuria. e-GFR, estimated glomerular filtration rate. IFTA, interstitial fibrosis and tubular atrophy. A two-tailed *p* < 0.05 was considered statistically significant.

**Table 1 jcm-12-00088-t001:** Baseline clinical findings in ESRD and non-ESRD groups.

	Total (*n* = 199)	Non-ESRD Group (*n* = 86)	ESRD Group (*n* = 113)	*p*
Gender (male)	58 (29.1)	27 (31.4)	31 (27.4)	0.651
Age (years)	51.49 ± 9.12	52.22 ± 9.81	50.93 ± 8.57	0.372
Duration of diabetes (m)	96 (36–132)	84 (36–120)	120 (36–144)	0.066
Cigarette smoking (%)	100 (50.3)	41 (47.7)	59 (52.2)	0.623
Hypertension (%)	180 (90.5)	80 (93)	100 (88.5)	0.405
SBP (mmHg)	149.94 ± 23.52	150.17 ± 24.41	149.77 ± 22.92	0.906
DBP (mmHg)	87.96 ± 13.57	89.64 ± 14.09	86.68 ± 13.08	0.132
e-GFR (mL/min/1.73 m^2^)	51.31 (36.15–81.54)	61.15 (45.28–93.30)	43.4 (27.57–59.74)	<0.001 *
Serum creatinine (umol/L)	139 (97–187)	108.5 (74.5–145.7)	156 (115–245.8)	<0.001 *
Stage 1,2,3a,3b,4,5 CKD (KDIGO)	33/42/47/38/32/7	19/13/21/17/15/1	14/29/26/21/17/6	0.157
Cys-C (mg/L)	1.72 (1.38–2.23)	1.41 (1.13–1.81)	1.93 (1.58–2.65)	<0.001 *
BUN (mmol/L)	8.2 (6.30–12.30)	7 (5.80–10.74)	9.31 (7.05–13.99)	<0.001 *
Uric acid (umol/L)	378 (327–428)	379 (320.5–418.5)	377.2 (329.5–435)	0.898
Total cholesterol (mmol/L)	5.24 (4.45–6.46)	5.17 (4.42–6.20)	5.24 (4.48–6.49)	0.448
HDL-C (mmol/L)	1.29 (1.04–1.59)	1.16 (1.05–1.5)	1.31 (1.04–1.67)	0.320
LDL-C (mmol/L)	3.13 (2.47–4.11)	3.07 (2.45–3.85)	3.14 (2.51–4.16)	0.686
Triglycerides (mmol/L)	1.86 (1.35–2.45)	1.92 (1.4–2.42)	1.83 (1.27–2.53)	0.526
Albumin (g/L)	30.04 ± 6.34	31.64 ± 6.21	28.83 ± 6.20	<0.001 *
Glucose (mmol/L)	7.64 (5.5–10.16)	7.93 (6.43–10.21)	7.59 (5.13–10.18)	0.329
HbA1c (%)	7.35 (6.3–8.6)	7.6 (6.7–8.8)	7.1 (6.0–8.4)	0.014 *
Proteinuria (g/24 h)	6.78 (4.8–9.75)	6.34 (4.46–9.32)	7.14 (5.01–10.17)	0.145
Hematuria (%)	145 (72.7)	62 (72.1)	83 (73.5)	0.958
Hemoglobin (g/L)	111 (98–129)	124 (109–143.5)	106 (90–116)	<0.001 *
Calcium (mmol/L)	2.08 ± 0.16	2.11 ± 0.14	2.06 ± 0.18	0.043 *
Phosphorus (mmol/L)	1.26 ± 0.27	1.22 ± 0.25	1.29 ± 0.29	0.062
Use of ACEI/ARB (%)	180 (90.5)	80 (93.0)	100 (88.5)	0.405
Use of Statin (%)	122 (61.3)	53 (61.6)	69 (61.1)	0.935
Use of Insulin (%)	149 (74.9)	56 (65.1)	95 (82.3)	0.008 *

ESRD, end-stage renal disease. SBP, systolic pressure. DBP, diastolic blood pressure. e-GFR, estimated glomerular filtration rate. Cys-C, Cystatin C. BUN, blood urea nitrogen. HDL-C, High-Density Lipoprotein Cholesterol. LDL-C, Low-Density Lipoprotein Cholesterol. ACEI/ARB, Angiotensin Converting Enzyme Inhibitors/Angiotensin Receptor Blocker. Data are presented as the mean ± standard, median and interquartile range, or percentages. * A two-tailed *p* < 0.05 was considered statistically significant.

**Table 2 jcm-12-00088-t002:** Baseline pathological findings in ESRD and non-ESRD groups.

Pathological Lesions	Total (*n* = 199)	Non-ESRD Group (*n* = 86)	ESRD Group (*n* = 113)	*p*
Glomerular class				<0.001 *
I	2 (1.00)	2 (2.32)	0 (0)	
IIa	28 (14.07)	21 (24.42)	7 (6.19)	
IIb	23 (11.56)	11 (12.79)	12 (10.62)	
III	108 (54.27)	38 (44.19)	70 (61.95)	
IV	38 (19.10)	14 (16.28)	24 (21.24)	
IFTA				0.047 *
0	1 (0.50)	1 (1.16)	0 (0)	
1	71 (35.68)	38 (44.19)	33 (29.20)	
2	96 (48.24)	34 (39.53)	62 (54.87)	
3	31 (15.58)	13 (15.12)	18 (15.93)	
Interstitial inflammation				0.003 *
0	8 (4.02)	7 (8.14)	1 (0.89)	
1	133 (66.83)	62 (72.09)	71 (62.83)	
2	58 (29.15)	17 (19.77)	41 (36.28)	
Arteriolar hyalinosis				0.153
0	13 (6.53)	7 (8.14)	6 (5.31)	
1	99 (49.75)	48 (55.81)	51 (45.13)	
2	87 (43.72)	31 (36.05)	56 (49.56)	

IFTA, interstitial fibrosis and tubular atrophy. Data are presented as percentages. * A two-tailed *p* < 0.05 was considered statistically significant.

**Table 3 jcm-12-00088-t003:** Baseline immunofluorescence findings in ESRD and non-ESRD groups.

Pathological Immunofluorescence	Total (*n* = 199)	Non-ESRD Group (*n* = 86)	ESRD Group (*n* = 113)	*p*
IgG (%)	30 (15.08)	6 (6.98)	24 (21.24)	0.010 *
IgM (%)	44 (22.11)	12 (13.95)	32 (28.32)	0.025 *
IgA (%)	15 (7.54)	5 (5.81)	10 (8.85)	0.594
C3 (%)	33 (16.58)	6 (6.98)	27 (23.89)	0.003 *
C4 (%)	20 (10.05)	4 (4.65)	16 (14.16)	0.049 *
C1q (%)	26 (13.07)	5 (5.81)	21 (18.58)	0.015 *

IgG, immunoglobin G. IgM, immunoglobin M. IgA, immunoglobin A. C3, complement C3. C4, complement C4. C1q, complement C1q. Data are presented as percentages. * A two-tailed *p* < 0.05 was considered statistically significant.

**Table 4 jcm-12-00088-t004:** Correlations between the histopathological and clinical findings.

	Glomerular Class	IFTA	Interstitial Inflammation	Arteriolar Hyalinosis	Albumin (g/L)	e-GFR	Hemoglobin (g/L)	Proteinuria (g/24 h)	ESRD
Glomerular class	1								
IFTA	0.450 **	1							
Interstitial inflammation	0.241 **	0.508 **	1						
Arteriolar hyalinosis	0.207 **	0.211 **	0.069	1					
Albumin (g/L)	−0.072	−0.062	−0.079	−0.048	1				
e-GFR	−0.367 **	−0.502 **	−0.484 **	−0.144 *	0.085	1			
Hemoglobin (g/L)	−0.333 **	−0.180 *	−0.180 *	−0.093	0.363 **	0.413 **	1		
Proteinuria (g/24 h)	0.122	0.128	0.099	0.073	−0.389 **	−0.168 *	−0.174 *	1	
ESRD	0.228 **	0.136	0.219 **	0.137	−0.253 **	−0.359 **	−0.437 **	0.103	1

** Correlation is significant at the 0.01 level (2-tailed). * Correlation is significant at the 0.05 level (2-tailed).

**Table 5 jcm-12-00088-t005:** Risk factors for ESRD identified by univariate Cox analysis in DKD patients with nephrotic range proteinuria.

Factors	HR	95% Cl	*p*
univariate			
Gender	0.946	0.625–1.432	0.794
Age (years)	0.991	0.971–1.012	0.401
Duration of diabetes (m)	1.003	1.000–1.005	0.042 *
Albumin (g/L)	0.942	0.914–0.970	<0.001 *
e-GFR (mL/min/1.73 m^2^)	0.971	0.962–0.979	<0.001 *
HbA1c (%)	0.909	0.819–1.010	0.075
Hemoglobin (g/L)	0.973	0.965–0.981	<0.001 *
Proteinuria (g/24 h)	1.052	1.015–1.091	0.006 *
Use of ACEI/ARB	0.525	0.356–0.774	0.001 *
Use of Statin	1.018	0.696–1.488	0.927
Use of Insulin	2.028	1.248–3.296	0.004 *
Glomerular damage	2.480	1.503–4.091	<0.001 *
IFTA	1.913	1.260–2.903	0.002 *
Interstitial inflammation	2.170	1.470–3.203	<0.001 *
Arteriolar hyalinosis	1.363	0.940–1.975	0.102

e-GFR, estimated glomerular filtration rate. IFTA, interstitial fibrosis and tubular atrophy. * A two-tailed *p* < 0.05 was considered statistically significant.

## Data Availability

The data presented in this study are available on request from the corresponding author. The data are not publicly available due to privacy restrictions.

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
