# Peer review of "Whether Renal Pathology Is an Independent Predictor for End-Stage Renal Disease in Diabetic Kidney Disease Patients with Nephrotic Range Proteinuria: A Biopsy-Based Study"

_jcm, 2022, doi:10.3390/jcm12010088_

Round 1

Reviewer 1 Report

The authors have clearly presented their data and explained their approach as well as results. However, there is a need to analyse the gender differences that the data might suggest. With various recent studies and analysis that have been done, people in the field of diabetic kidney disease have identified that the disease progresses different in the male and female. So it is important to address this issue here. The title of the manuscript does not justify the conclusions or the hypothesis. The use of renal biopsy is usually not the case for the diagnosis of DKD or CKD. In the nephrology community, the proteinuria and GFR are most commonly used to diagnosis DKD as suggested by this studies conclusion. 

Reviewer 2 Report

The authors have carried out a single-center retrospective study to correlate the development of ESRD in diabetic patients with renal pathology in order to determine if the latter could be used as a predictor/indicator of adverse renal outcomes. The study is well designed and the authors do take multiple factors into account while setting their exclusion criteria. For this kind of study the sample size is relatively small (199 patients) and the authors do mentions this in the discussion as a limitation. The data is presented well and the authors have used the right methods for their correlation analyses. The discussion is well written and the authors present the drawbacks and limitations of their study given that it is a real world analysis. 

Minor comments:

For experts not in the field, an abbreviation table will be helpful and/or mentioning the full term at the first use (eg/ NDRD, Hb etc)

Reviewer 3 Report

In the present retrospective study, the authors investigated the prognostic factors of renal survival in DKD patients with nephrotic range proteinuria. Baseline pathological evaluation was used in the present study. Overall, the manuscript is well-written. I have a few comments below.

11   Does the author exclude patients with type 1 DM? In patients with type 2 DM, microvascular complications may be available during the diagnosis. Therefore, the definition ‘duration of diabetes’ may not be the right description.

22     Glycemia regulation is one of the most important parameters affecting microvascular complications in patients with diabetes. Many lab parameters were provided in Table 1. But I could not see hemoglobin A1c level, which may affect DKD progression. Moreover, I was also wondering about the association between baseline A1c level and renal outcomes if the authors have the data.

33.    The quality of the images is not good in Figure 2. It may be improved.

Round 2

Reviewer 1 Report

Thank you for answering the comments and providing evidence as well as references to support your work